# Uterine Septum with or without Hysteroscopic Metroplasty: Impact on Fertility and Obstetrical Outcomes—A Systematic Review and Meta-Analysis of Observational Research

**DOI:** 10.3390/jcm11123290

**Published:** 2022-06-08

**Authors:** Marco Noventa, Giulia Spagnol, Matteo Marchetti, Carlo Saccardi, Giulio Bonaldo, Antonio Simone Laganà, Francesco Cavallin, Alessandra Andrisani, Guido Ambrosini, Salvatore Giovanni Vitale, Luis Alonso Pacheco, Sergio Haimovich, Attilio Di Spiezio Sardo, Jose Carugno, Marco Scioscia, Simone Garzon, Stefano Bettocchi, Giovanni Buzzaccarini, Roberto Tozzi, Amerigo Vitagliano

**Affiliations:** 1Unit of Gynecology and Obstetrics, Department of Women and Children’s Health, University of Padua, 35100 Padua, Italy; giuliaspagnol.ts@gmail.com (G.S.); matteomarchetti91@gmail.com (M.M.); carlo.saccardi@unipd.it (C.S.); bonaldo.giulio@gmail.com (G.B.); alessandra.andrisani@unipd.it (A.A.); guido.ambrosini@unipd.it (G.A.); giovanni.buzzaccarini@gmail.com (G.B.); roberto.tozzi@unipd.it (R.T.); amerigo.vitagliano@gmail.com (A.V.); 2Unit of Gynecology Oncology, Department of Health Promotion, Mother and Child Care, Internal Medicine and Medical Specialities (PROMISE), University of Palermo, 90127 Palermo, Italy; antoniosimone.lagana@unipa.it; 3Independent Researcher, 36020 Solagna, Italy; cescocava@yahoo.it; 4Obstetrics and Gynecology Unit, Department of General Surgery and Medical Surgical Specialties, University of Catania, 95124 Catania, Italy; vitalesalvatore@hotmail.com; 5Endoscopy Unit, Centro Gutenberg, 29003 Málaga, Spain; luisalonso2@gmail.com; 6Hillel Yaffe Medical Center, Technion-Israel Technology Institute, Hadera 38100, Israel; sergio.haimovich@gmail.com; 7Department of Public Health, School of Medicine, University of Naples Federico II, 80138 Naples, Italy; attiliodispiezio@libero.it; 8Minimally Invasive Gynecology Unit, Obstetrics, Gynecology and Reproductive Sciences Department, Miller School of Medicine, University of Miami, Miami, FL 33136, USA; tonycarugno@yahoo.com; 9Unit of Gynecology, Mater Dei Hospital, 70125 Bari, Italy; marco.scioscia@gmail.com; 10Department of Obstetrics and Gynaecology, AOUI Verona, University of Verona, 37126 Verona, Italy; simone.garzon@univr.it; 11Inter-Departmental Project Unit of Minimal-Invasive Gynecological Surgery, Policlinico of Bari, University of Bari Aldo Moro, 70121 Bari, Italy; stefano.bettocchi@uniba.it

**Keywords:** uterine septum, metroplasty, pregnancy rate, live birth rate, spontaneous abortion, preterm labour, infertility, recurrent miscarriage

## Abstract

Objective: we performed a systematic review/meta-analysis to evaluate the impact of septate uterus and hysteroscopic metroplasty on pregnancy rate-(PR), live birth rate-(LBR), spontaneous abortion-(SA) and preterm labor (PL) in infertile/recurrent miscarriage-(RM) patients. Data sources: a literature search of relevant papers was conducted using electronic bibliographic databases (Medline, Scopus, Embase, Science direct). Study eligibility criteria: we included in this meta-analysis all types of observational studies that evaluated the clinical impact of the uterine septum and its resection (hysteroscopic metroplasty) on reproductive and obstetrics outcomes. The population included were patients with a diagnosis of infertility or recurrent pregnancy loss. Study appraisal and synthesis methods: outcomes were evaluated according to three subgroups: (i) Women with untreated uterine septum versus women without septum (controls); (ii) Women with treated uterine septum versus women with untreated septum (controls); (iii) Women before and after septum removal. Odds ratios (OR) with 95% confidence intervals (CI) were calculated for the outcome measures. A *p*-value < 0.05 was considered statistically significant. Subgroup analysis was performed according to the depth of the septum. Sources of heterogeneity were explored by meta-regression analysis according to specific features: assisted reproductive technology/spontaneous conception, study design and quality of papers included Results: data from 38 studies were extracted. (i) septum versus no septum: a lower PR and LBR were associated with septate uterus vs. controls (OR 0.45, 95% CI 0.27–0.76; *p* < 0.0001; and OR 0.21, 95% CI 0.12–0.39; *p* < 0.0001); a higher proportion of SA and PL was associated with septate uterus vs. controls (OR 4.29, 95% CI 2.90–6.36; *p* < 0.0001; OR 2.56, 95% CI 1.52–4.31; *p* = 0.0004). (ii) treated versus untreated septum: PR and PL were not different in removed vs. unremoved septum(OR 1.10, 95% CI 0.49–2.49; *p* = 0.82 and OR 0.81, 95% CI 0.35–1.86; *p* = 0.62); a lower proportion of SA was associated with removed vs. unremoved septum (OR 0.47, 95% CI 0.21–1.04; *p* = 0.001); (iii) before-after septum removal: the proportion of LBR was higher after the removal of septum (OR 49.58, 95% CI 29.93–82.13; *p* < 0.0001) and the proportion of SA and PL was lower after the removal of the septum (OR 0.02, 95% CI 0.02–0.04; *p* < 0.000 and OR 0.05, 95% CI 0.03–0.08; *p* < 0.0001) Conclusions: the results show the detrimental effect of the uterine septum on PR, LBR, SA and PL. Its treatment reduces the rate of SA.

## 1. Introduction

Congenital uterine anomalies (CUAs) are estimated to affect 1% to 4% of the general fertile population [1], although the reported prevalence is significantly higher (8.5–12%) in patients with infertility and recurrent pregnancy loss [2]. The uterine septum is the most common CUA, representing 35% of all of the diagnosed malformations [3,4].

The definition of uterine septum is still a subject of debate. Two main classification systems were developed: the ASRM (AFS) classification initially published in 1988, which was subsequently modified in 2016, and the ESHRE-ESGE classification [5,6,7]. It is accepted that both classification systems have some limitations. In particular, ESHRE-ESGE classification may overestimate the prevalence of septate uterus while that of ASRM may underestimate it, leaving in a gray-zone most of the uteri that could considered as septate. To address this important clinical dilemma, a novel cut-off of >10 mm indentation depth was proposed by the Congenital Uterine Malformation by Experts (CUME) in 2018 [8].

Whether the uterine septum increases the risk of reproductive failure is still uncertain. On the one hand, several observational studies reported an association between the uterine septum and obstetrical complications including recurrent spontaneous abortions (both in the first and second trimester), preterm delivery, intrauterine growth restriction (IUGR) and placental abruption [3,9,10,11]. Moreover, some authors suggest that hysteroscopic metroplasty of the septum can reduce miscarriage rates and improve the obstetrical outcomes in patients with a history of recurrent miscarriage [1,12,13]. On the other hand, a few studies support a causal effect of the uterine septum on infertility, [13,14,15,16], as demonstrated by an improvement of implantation rates after septum removal [12,17,18,19].

According to recent studies, the uterine septum may impair the embryo implantation and development through both molecular and mechanical mechanisms. Lower expression of the homeobox protein Hox-A10 (HOXA10) and Vascular Endothelial Growth factor (VEGF) receptor genes, lower number of glandular and ciliated cells in the endometrial lining of the intrauterine septum and uncoordinated uterine contractility (due to increased content of muscle tissue within the septum) are supposed to be the main factors [20,21].

An important effort in summarizing the published data on CUAs and reproductive issues was given by Venetis et al. [13]. Their review found an association among the presence of CUAs and a reduction in clinical pregnancy rates, an increase in the first- or second-trimester spontaneous abortion rates, and preterm delivery rates. However, the study reported by Venetis et al. included all types of uterine anomalies, which limits its clinical significance when limited to the management of the uterine septum [13]. The first randomized controlled trial on hysteroscopic septum resection in women with reproductive disorders was recently published [22]. In this study, Rikken et al. found no improvements in reproductive outcomes from the intervention, thereby questioning any rationale supporting surgical management. Nevertheless, the study by Rikken et al. has been strongly criticized, recognizing several limitations that preclude the interpretation of the data, including the small sample size (*n* = 68 women) and heterogeneity in terms of patients’ characteristics and reproductive disorders. Therefore, the debate on the impact of the uterine septum and its surgical correction on reproductive outcomes is still open.

Over the above-described controversial background, the aim of our systematic review was to summarize evidence from observational studies on the impact of the septate uterus on pregnancy rate, live birth rate, first- or second-trimester spontaneous abortion rates and preterm delivery rates in women suffering from infertility or recurrent miscarriage. We also evaluated the impact of hysteroscopic metroplasty on the same clinical outcomes.

## 2. Material and Methods

### 2.1. Study Design and Protocol Registration

This meta-analysis was conducted following Meta-analysis of Observational Studies in Epidemiology (MOOSE) guidelines [23]. The research study, eligibility criteria, study selection and data extraction process were defined a priori. The review protocol was recorded on the international prospective register of systematic reviews PROSPERO (registration ID: CRD42020196157) before starting the literature search.

### 2.2. Eligibility Criteria

We included in this meta-analysis all types of observational studies that evaluated the clinical impact of the uterine septum and its resection (hysteroscopic metroplasty) on reproductive and obstetrics outcomes. We excluded from the analysis narrative or systematic reviews and case reports.

The definition of uterine septum, the diagnostic criteria (according to guidelines) and the diagnostic methods varied among papers; a detailed description is reported in Table 1, Table 2 and Table 3.

The inclusion criteria applied in this meta-analysis were the following:Type of study: Case-control studies, cohort studies or case series.Period of publication: no restriction.Language: English.Participants: Women diagnosed with infertility or recurrent pregnancy loss (categories were intended as classified by authors). See Table 1, Table 2 and Table 3 for details.Comparators: (i) Women with untreated uterine septum versus women without septum (controls); (ii) Women with treated uterine septum versus women with untreated septum (controls); (iii) Women before and after septum removal.Outcomes: Pregnancy rate; live birth rate; spontaneous abortion rate; preterm delivery rate.Outcome definitions: Pregnancy rate (PR—defined as the presence of a gestational sac on transvaginal ultrasound; Live birth rate (LBR—defined as the delivery of one or more living and viable infants). Spontaneous abortion rate (SA—defined as fetal loss prior to the completed 23th week of gestation; Preterm labour (PL—defined as a delivery before 37 weeks of gestation).

### 2.3. Information Sources and Search Strategies

A literature search of relevant papers was conducted using the electronic bibliographic databases (Medline, Scopus, Embase, Science direct, Cochrane library, Clinicaltrials.gov, Cochrane Central Register of Controlled Trials, EU Clinical Trials Register and World Health Organization International Clinical Trials Registry Platform). The strategies for electronic search were the following combined search: Uterine septum OR Septate uterus OR Metroplasty AND Pregnancy rate OR Live birth rate OR Spontaneous miscarriage OR infertility OR preterm delivery.

### 2.4. Study Selection

Titles and/or abstracts of studies retrieved using the electronic search strategy and those from additional sources were screened independently by two review authors (GS, MM) to identify studies that potentially meet the inclusion criteria outlined above. The full text of these potentially eligible studies was retrieved and independently assessed for eligibility by other two review team members (MN, GB). Any disagreement over the eligibility of a study was resolved through discussion with a third external collaborator (AV). A standardized, pre-piloted form was used to extract data from the included studies for assessment of study quality and evidence synthesis. According to this form, we extracted data about studies characteristics (design and time of the study), population (number of enrolled women, general characteristics, patient’s selection, definition of septum (ASRM, ESHRE-ESGE and CUME classification) [5,6,7,8], diagnostic technique (3D transvaginal ultrasound and/or hysteroscopy), depth of septum (partial or complete), metroplasty technique performed and specific outcomes including pregnancy rate and live birth rate [LBR from spontaneous conception or after assisted reproductive technology (ART)], spontaneous abortion and preterm labor.

A manual search of a reference list of included studies was also performed in order to avoid missing relevant data. We searched for published (full-text studies and meeting abstracts) and unpublished studies (i.e., for whom only a registered protocol was available) from the aforementioned electronic databases. The results were compared, and any disagreement was resolved by consensus.

### 2.5. Risk of Bias

Two reviewers (G.S. and G.B.) independently judged the methodological quality of the studies included in the meta-analysis. Outcome’s selection and measurement was assessed for each distinct outcome. Disagreements between the reviewers over the risk of bias of any study were solved by discussion with a third review author (M.N).

The risk of bias was be assessed for each study using the methodological index for non-randomized studies (MINORS) [59]. The tool consists in 12 items, the first eight are methodological items for non-randomized studies and the last four items are additional criteria to apply in the case of comparative study. Each item is scored as 0 (not reported), 1 (reported but inadequate) or 2 (reported and adequate). The global ideal score is 16 for non-comparative studies and 24 for comparative studies. For comparative studies we considered a paper of high quality with a score of 23–24, medium quality with a score of 21–22 and low quality with a score ≤20. For non-comparative studies we considered a paper of high quality with a score of 15–16, medium quality with a score of 13–14 and low quality with a score ≤ 12.

### 2.6. Statistical Analysis

The data analysis was performed by two authors (F.C. and M.N.) using “metafor” package for R software, version 4.0 (R Foundation for Statistical Computing, Vienna, Austria) [60,61]. A *p*-value less than 0.05 was considered statistically significant. Odds ratios (OR) with 95% confidence intervals (CI) were calculated for the outcome measures in each study. Meta-analyses were performed using random-effects models. Each outcome was analyzed independently. Heterogeneity was assessed using the I^2^ value and potential sources of heterogeneity were explored using meta-regression; an I^2^ value over 50% indicated substantial heterogeneity [62]. The risk of publication bias was assessed using funnel plots and the trim-and-fill method [63]. Of note, this method was used to suggest possible missing studies, but not to provide adjusted estimates (those resulting from a meta-analysis including the filled studies) [64].

Subgroup analysis was performed according to the depth of the septum (partial or complete) including the following population: women diagnosed with infertility and women diagnosed with recurrent pregnancy loss. Sources of heterogeneity were explored by meta-regression analysis according to the features of the included studies: assisted reproductive technology/spontaneous conception, study design (prospective/retrospective) and quality of papers included (low/medium/high). Sensitivity analyses including only medium and high-quality studies was also performed.

## 3. Results

### Study Selection

The electronic searches provided a total of 2390 citations: after the removal of 550 duplicate records, 1840 references remained. Of these, 1780 records were excluded after title/abstract screening (not relevant to the review). We examined the full text of 60 remaining manuscripts, and of these, 21 papers were excluded (four studies with overlapping publication were excluded, seven studies evaluating all CUA and did not analyze septate uteri, ten studies not evaluating hysteroscopic removal of uterine septa). Only one randomized trial was available [22]; however, considering that the design of all other trials was observational and that it would be a methodological mistake to aggregate data from observational and randomized studies together, we decided to exclude this last article from the aggregate analysis and to examine it in the discussion section. So, 38 observational papers were included in the meta-analysis. Ten manuscripts were included in the first section: infertile/recurrent miscarriage patients with uterine septum versus controls (no septum) [10,14,26,27,28,29,30,31,32,33]; nine manuscripts were included in the second section: infertile/recurrent miscarriage patients with treated uterine septum versus controls (untreated septum) [19,29,34,35,36,37,38,39,40]; and, finally, nineteen manuscripts were included in the third section: infertile/recurrent miscarriage patients before and after septum removal [12,41,42,43,44,45,46,47,48,49,50,51,52,53,54,55,56,57,58]. Figure 1 shows the flow chart summarizing literature identification and selection.

## 4. Research Findings

### 4.1. Uterine Septum vs. Controls (No Septum)

In the first section we included ten manuscripts for a total of 6182 patients (495 cases and 5687 controls). For details see Table 1 (study characteristics) [10,14,26,27,28,29,30,31,32,33]. 

### 4.2. Pregnancy Rate

PR was investigated in six studies [10,14,26,30,31,33]. A lower PR was associated with septate uterus vs. controls (OR 0.45, 95% CI 0.27 to 0.76; *p* < 0.0001) with low heterogeneity (I^2^ = 25%) (Figure 2A). After meta-regression analysis, spontaneous conception vs assisted conception (*p* = 0.75), study design (*p* = 0.26) and study quality (*p* = 0.69) did not change the effect size.

#### 4.2.1. Subgroup Analysis

In four studies, PR was lower in women with subseptate uterus vs. controls (OR 0.56, 95% CI 0.30 to 1.07; low heterogeneity I^2^= 0%) but did not reach statistical significance (*p* = 0.07) [10,14,26,33].

In three studies, PR was not different in women with complete septate uterus vs. subseptate uterus (OR 1.40, 95% CI 0.12 to 16.87; *p* = 0.79) with moderate heterogeneity (I^2^ = 56%) [10,14,33].

In three studies reporting data from infertile patients, PR was not different in women with septate uterus vs. controls (OR 0.58, 95% CI 0.15 to 2.16; *p* = 0.42) with substantial heterogeneity (I^2^ = 80%) [10,31,33].

In one study reporting data from recurrent abortion, PR was not different in septate uterus vs. controls (OR 0.39, 95% CI 0.10 to 1.54; *p* = 0.18) [30].

#### 4.2.2. Sensitivity Analysis

In four studies with medium/high quality, PR was not different in women with septate uterus vs. controls (OR 0.54, 95% CI 0.25 to 1.18; *p* = 0.12) with moderate heterogeneity (I^2^ = 63%). [10,26,31,33].

### 4.3. Live Birth Rate

LBR was investigated in four studies [10,14,31,32]. A lower LBR was associated with septate uterus vs. controls (OR 0.21, 95% CI 0.12 to 0.39; *p* < 0.0001) with small heterogeneity (I^2^= 19%) (Figure 2C). At meta-regression analysis, spontaneous conception vs assisted conception (*p* = 0.11) and study quality (*p* = 0.18) did not change the effect size.

#### 4.3.1. Subgroup Analysis

In two studies, a lower LBR was associated with subseptate uterus vs. controls (OR 0.24, 95% CI 0.08 to 0.68; *p* = 0.008) with low heterogeneity (I^2^= 0%) [10,14].

In two studies, LBR was not different in women with complete septate uterus vs. subseptate uterus (OR 0.61, 95% CI 0.22 to 1.67; *p* = 0.33) with low heterogeneity (I^2^ = 0%) [10,14].

In two studies reporting data from infertile patients, LBR was lower in women with septate uterus vs. controls (OR 0.10, 95% CI 0.03 to 0.31; *p* < 0.0001) with low heterogeneity (I^2^ = 0%) [10,31].

In one study reporting data from recurrent abortion, LBR was lower in women with septate uterus vs. controls (OR 0.33, 95% CI 0.17 to 0.64; *p* = 0.001) [32].

#### 4.3.2. Sensitivity Analysis

In three studies with medium/high quality, LBR was lower in women with septate uterus vs. controls (OR 0.15, 95% CI 0.07 to 0.31; *p* < 0.0001) with low heterogeneity (I^2^ = 0%) [14,31,32].

### 4.4. Spontaneous Abortions in I-II Trimesters

SA in I-II trimesters was investigated in 10 studies. A higher proportion of SA was associated with septate uterus vs. controls (OR 4.29, 95% CI 2.90 to 6.36; *p* < 0.0001) with moderate heterogeneity (I^2^= 43%) [10,14,26,27,28,29,30,31,32,33] (Figure 2B).

The influence of spontaneous conception vs assisted conception (*p* = 0.06) was not statistically significant, while septum classification (*p* = 0.43) and study quality (*p* = 0.41) did not change the effect size. Overall, the funnel plot was symmetrical and the trim-and-fill method by Duval and Tweedie did not suggest any missing studies (Appendix A).

#### 4.4.1. Subgroup Analysis

In five studies, SA was higher in women with subseptate uterus vs. controls (OR 4.40, 95% CI 2.94 to 6.57; *p* < 0.0001) with low heterogeneity (I^2^= 0%). 

In four studies, SA was not different in women with complete septate uterus vs. subseptate uterus (OR 0.87, 95% CI 0.44 to 1.71; *p* = 0.68) with low heterogeneity (I^2^ = 0%) [10,14,28,33].

In five studies reporting data from infertile patients, SA was higher in women with septate uterus vs. controls (OR 7.56, 95% CI 3.38 to 16.93; *p* < 0.0001) with moderate heterogeneity (I^2^ = 45%) [10,27,29,31,33].

In two studies reporting data from recurrent abortion, SA was higher in women with septate uterus vs. controls (OR 2.29, 95% CI 1.32 to 3.99; *p* = 0.003) [30,32].

#### 4.4.2. Sensitivity Analysis

In five studies with medium/high quality, SA was higher in women with septate uterus vs. controls (OR 5.13, 95% CI 3.33 to 7.92; *p* < 0.0001) with low heterogeneity (I^2^ = 0%) [10,14,26,31,33].

#### 4.4.3. Spontaneous Abortion during the First Trimester of Pregnancy

SA during the first trimester of pregnancy was investigated in five studies [14,26,28,32,33]. A higher proportion of SA was associated with septate uterus vs. controls (OR 3.15, 95% CI 1.59 to 6.2; *p* = 0.001) with substantial heterogeneity (I^2^= 78%). The influence of study design was not statistically significant (*p* = 0.06), while study quality (*p* = 0.21) did not change the effect size.

In four studies, SA was higher in women with subseptate uterus vs. controls (OR 4.36, 95% CI 2.51 to 7.57; *p* < 0.0001) with moderate heterogeneity (I^2^= 24%) [14,26,28,33].

In three studies, SA was not different in women with complete septate uterus vs. subseptate uterus (OR 0.92, 95% CI 0.45 to 1.87; *p* = 0.81) with low heterogeneity (I^2^ = 0%) [14,28,33].

In three studies with medium/high quality, SA was higher in women with septate uterus vs. controls (OR 5.45, 95% CI 3.36 to 8.85; *p* < 0.0001) with low heterogeneity (I^2^ = 0%) [14,26,33].

#### 4.4.4. Spontaneous Abortions during the Second Trimester of Pregnancy

SA during the second trimester of pregnancy was investigated in four studies. A higher proportion of SA was associated with septate uterus vs. controls (OR 2.69, 95% CI 1.11 to 6.52; *p* = 0.03) with moderate heterogeneity (I^2^= 56%) [14,26,28,32]. The influence of study quality was not statistically significant (*p* = 0.06).

In two studies, SA was not different in women with subseptate uterus vs. controls (OR 2.24, 95% CI 0.51 to 9.83; *p* = 0.28) with moderate heterogeneity (I^2^ = 60%) [26,28].

In one study, SA was not different in women with complete septate uterus vs. subseptate uterus (OR 1.04, 95% CI 0.29 to 3.67; *p* = 0.95) [28].

In three studies with medium/high quality, SA was not different in women with septate uterus vs. controls (OR 0.79, 95% CI 0.21 to 2.92; *p* = 0.72) with low heterogeneity (I^2^ = 0%) [14,26,28].

### 4.5. Preterm Labor

Preterm labor was investigated in three studies [14,26,28]. A higher proportion of preterm labor was associated with septate uterus vs. controls (OR 2.56, 95% CI 1.52 to 4.31; *p* = 0.0004) with low heterogeneity (I^2^= 0%) (Figure 2D). An analysis of moderators could not be performed due to the small sample size.

#### 4.5.1. Subgroup Analysis

In three studies, preterm labor was higher in women with subseptate uterus vs. controls (OR 2.98, 95% CI 1.05 to 8.47; *p* = 0.04) with low heterogeneity (I^2^ = 0%) [14,26,28].

In two studies, preterm labor was not different in women with complete septate uterus vs. subseptate uterus (OR 0.80, 95% CI 0.24 to 2.69; *p* = 0.72) with moderate heterogeneity (I^2^ = 37%) [14,28].

#### 4.5.2. Sensitivity Analysis

In two studies with medium/high quality, preterm labor was not different in women with septate uterus vs. controls (OR 2.87, 95% CI 0.96 to 8.60; *p* = 0.06) with moderate heterogeneity (I^2^ = 41%) [14,26].

### 4.6. Treated Uterine Septum versus Untreated Uterine Septum

In the second section we analyzed nine manuscripts including a total of 1053 patients (603 cases and 450 controls). Table 2 (study features) [19,29,34,35,36,37,38,39,40]. 

### 4.7. Pregnancy Rate

PR was investigated in six studies [19,35,36,37,38,39]. PR was not different in in women who had the uterine septum removed vs. not removed (OR 1.10, 95% CI 0.49 to 2.49; *p* = 0.82) with moderate heterogeneity (I^2^= 66%) (Figure 3A). Study design (*p* = 0.74) and study quality (*p* = 0.20) did not change the effect.

#### 4.7.1. Subgroup Analysis

In three studies reporting data from infertile patients, PR was not different in women with septate uterus vs. controls (OR 1.66, 95% CI 0.48 to 5.73; *p* = 0.42) with moderate heterogeneity (I^2^ = 58%) [19,36,39].

In four studies reporting data from recurrent abortion, PR was not different in women who had the uterine septum removed vs. not removed (OR 1.22, 95% CI 0.51 to 2.92; *p* = 0.66) with moderate heterogeneity (I^2^ = 47%) [35,37,38,39].

#### 4.7.2. Sensitivity Analysis

All six studies were judged to be of medium quality, thus the sensitivity analysis overlapped with the complete analysis [19,35,36,37,38,39].

### 4.8. Live Birth Rate

LBR was investigated in eight studies. [19,34,35,36,37,38,39,40]. A higher LBR was associated with removed vs. unremoved septate uterus (OR 3.07, 95% CI 1.22 to 7.73; *p* = 0.02) with substantial heterogeneity (I^2^ = 76%) (Figure 3B). The effect size was enhanced in higher quality studies (OR 1.40, 95% CI 1.01 to 1.94; *p* = 0.04; I^2^ = 67%), while the study design (*p* = 0.98) did not change the effect size.

#### 4.8.1. Subgroup Analysis

In three studies reporting data from infertile patients, LBR was not different in women with septate uterus vs. controls (OR 2.03, 95% CI 0.42 to 9.73; *p* = 0.38) with moderate heterogeneity (I^2^ = 34%) [19,36,39].

In six studies reporting data from recurrent abortion, LBR was not different in women who had the uterine septum removed vs. not removed (OR 2.96, 95% CI 0.96 to 9.15; *p* = 0.06) with substantial heterogeneity (I^2^ = 78%) [35,36,37,38,39,40].

#### 4.8.2. Sensitivity Analysis

In seven studies with medium/high quality, LBR was higher in women who had the septum removed vs. not removed (OR 3.87, 95% CI 1.47 to 10.20; *p* = 0.006) with substantial heterogeneity (I^2^ = 74%) [19,34,35,36,37,38,39]

### 4.9. Spontaneous Abortions during the First and Second Trimesters

SA during the first and second trimesters was investigated in nine studies [19,29,34,35,36,37,38,39,40]. SA seemed lower in women who had their uterine septum removed vs. not removed (OR 0.47, 95% CI 0.21 to 1.04; substantial heterogeneity I^2^= 80%) but did not achieved statistical significance (*p* = 0.06) (Figure 3C). Population (*p* = 0.15), study design (*p* = 0.47) and study quality (*p* = 0.09) did not change the effect size.

#### 4.9.1. Subgroup Analysis

In four studies reporting data from infertile patients, SA was not different in women with septate uterus vs. controls (OR 0.67, 95% CI 0.18 to 2.57; *p* = 0.56) with substantial heterogeneity (I^2^ = 72%) [19,29,36,39].

In six studies reporting data from recurrent abortion, SA was not different in women who had their uterine septum removed vs. not removed (OR 0.40, 95% CI 0.13 to 1.27; *p* = 0.12) with substantial heterogeneity (I^2^ = 79%) [35,36,37,38,39].

#### 4.9.2. Sensitivity Analysis

In eight studies with medium/high quality, SA was lower in women who had their uterine septum removed vs. not removed (OR 0.40, 95% CI 0.17 to 0.96; *p* = 0.04) with substantial heterogeneity (I^2^ = 81%) [19,29,34,35,36,37,38,39].

### 4.10. Preterm Labor

Preterm labor was investigated in eight studies [19,34,35,36,37,38,39]. Preterm labor was not different in women who had their uterine septum removed vs. not removed (OR 0.81, 95% CI 0.35 to 1.86; *p* = 0.62) with moderate heterogeneity (I^2^= 32%) (Figure 3D). The influence of study design was close not statistically significant (*p* = 0.07), while study quality (*p* = 0.19) did not change the effect size.

#### 4.10.1. Subgroup Analysis

In three studies reporting data from infertile patients, preterm labor was not different in women who had their uterine septum removed vs. not removed (OR 1.05, 95% CI 0.34 to 3.21; *p* = 0.94) with low heterogeneity (I^2^ = 0%) [19,36,39].

In six studies reporting data from recurrent abortion, preterm labor was not different in women who had their uterine septum removed vs. not removed (OR 0.74, 95% CI 0.26 to 2.15; *p* = 0.58) with moderate heterogeneity (I^2^ = 36%) [35,36,37,38,39,40].

#### 4.10.2. Sensitivity Analysis

In seven studies with medium/high quality, preterm labor was not different in women who had their uterine septum removed vs. not removed (OR 0.72, 95% CI 0.30 to 1.74; *p* = 0.46) with moderate heterogeneity (I^2^ = 36%) [19,34,35,36,37,38,39].

## 5. Before and after Septum Removal

In the third section we analyzed nineteen manuscripts including a total of 1920 patients. Table 3 (study features). [12,41,42,43,44,45,46,47,48,49,50,51,52,53,54,55,56,57,58]. 

### 5.1. Live Birth Rate

LBR was investigated in 14 studies [12,45,46,47,48,49,50,52,53,54,55,56,57,58]. LBR was higher after vs. before the removal of uterine septum (OR 49.58, 95% CI 29.93 to 82.13; *p* < 0.0001) with moderate heterogeneity (I^2^= 64%) (Figure 4A). Study design (*p* = 0.96) and study quality (*p* = 0.18) did not change the effect size.

Overall, the funnel plot was asymmetrical and the trim-and-fill method by Duval and Tweedie suggested four missing studies (with sample sizes of 50, 185, 255 and 263 pregnancies, and ORs of 3.67, 5.31, 2.15 and 1.23, respectively) in the left area of the plot (Appendix A).

#### 5.1.1. Subgroup Analysis

In four studies reporting data from infertile patients, LBR was higher after vs. before the removal of the uterine septum (OR 20.06, 95% CI 2.53 to 158.85; *p* = 0.005) with moderate heterogeneity (I^2^ = 43%) [12,47,53,56].

In six studies reporting data from recurrent abortion, LBR was higher after vs. before the removal of the uterine septum (OR 41.48, 95% CI 16.01 to 107.48; *p* < 0.0001) with substantial heterogeneity (I^2^ = 81%) [12,46,47,52,53,57].

#### 5.1.2. Sensitivity Analysis

In six studies with medium/high quality, PR was higher after vs. before the removal of the uterine septum (OR 68.08, 95% CI 36.95 to 125.41; *p* < 0.0001) with moderate heterogeneity (I^2^ = 58%) [12,49,50,52,54,58].

### 5.2. Spontaneous Abortions during the First and Second Trimesters of Pregnancy

SA during the first and second trimesters of pregnancy was investigated in 19 studies [12,41,42,43,44,45,46,47,48,49,50,51,52,53,54,55,56,57,58]. The proportion of abortions was lower after vs. before the removal of the uterine septum (OR 0.02, 95% CI 0.02 to 0.04; *p* < 0.0001) with moderate heterogeneity (I^2^ = 64%) (Figure 4B). Study design (*p* = 0.99) and study quality (*p* = 0.11) did not change the effect size.

Overall, the funnel plot was asymmetrical and the trim-and-fill method by Duval and Tweedie suggested two missing studies (with sample sizes of 255 and 263 pregnancies, and ORs 0.42 and 0.94, respectively) in the right area of the plot (Appendix A).

#### 5.2.1. Sub-Analyses

In five studies reporting data from infertile patients, SA was lower after vs. before the removal of the uterine septum (OR 0.04, 95% CI 0.01 to 0.09; *p* < 0.0001) with moderate heterogeneity (I^2^ = 23%).

In seven studies reporting data from recurrent abortion, SA was lower after vs. before the removal of the uterine septum (OR 0.03, 95% CI 0.01 to 0.06; *p* < 0.0001) with substantial heterogeneity (I^2^ = 78%).

#### 5.2.2. Sensitivity Analysis

In six studies with medium/high quality, SA was lower after vs. before the removal of the uterine septum (OR 0.02, 95% CI 0.01 to 0.03; *p* < 0.0001) with moderate heterogeneity (I^2^ = 50%).

Excluding data from three abstracts, SA was lower after vs. before the removal of the uterine septum (OR 0.02, 95% CI 0.02 to 0.03; *p* < 0.0001) with moderate heterogeneity (I^2^ = 54%).

### 5.3. Preterm Labor

Preterm labor was investigated in ten studies [12,41,42,43,44,46,50,51,54,55]. The proportion of preterm labor was lower after vs. before the removal of the uterine septum (OR 0.05, 95% CI 0.03 to 0.08; *p* ≤ 0.0001) with low heterogeneity (I^2^ = 0%) (Figure 4C). Study quality (*p* = 0.38) did not change the effect size.

Overall, the funnel plot was slightly asymmetrical and the trim-and-fill method by Duval and Tweedie suggested one missing study (with sample sizes of 29 pregnancies and OR of 0.01) in the left area of the plot (Appendix A).

#### 5.3.1. Subgroup-Analyses

In two studies reporting data from infertile patients, preterm labor was not different after vs. before the removal of the uterine septum (OR 11.61, 95% CI 0.05 to 2702.21; *p* = 0.38) with substantial heterogeneity (I^2^ = 84%).

In three studies reporting data from recurrent miscarriages, preterm labor was lower after vs. before the removal of the uterine septum (OR 0.03, 95% CI 0.01 to 0.09; *p* < 0.0001) with low heterogeneity (I^2^ = 0%).

#### 5.3.2. Sensitivity Analysis

In two studies with medium/high quality, preterm labor was lower after vs. before the removal of the uterine septum (OR 0.03, 95% CI 0.02 to 0.08; *p* < 0.0001) with low heterogeneity (I^2^ = 0%).

Excluding data from three abstracts, preterm labor was lower after vs. before the removal of the uterine septum (OR 0.04, 95% CI 0.02 to 0.08; *p* < 0.0001) with low heterogeneity (I^2^ = 0%).

## 6. Surgical Complications

Considering the nine papers of the section “Treated uterine septum versus controls (untreated septum)” a total of eleven complications were reported (*n* = 11 out of 603 procedures; complication rate of 1.8%). The reported complications were five uterine perforations (only one requiring laparoscopic suturing); five abnormal uterine bleeding and only one fluid overload syndrome (Appendix A).

Considering the nineteen papers of the section “Before and after septum removal” a total of 30 complications were reported (*n* = 30 out of 1920 procedures; complication rate of 1.5%). The reported complications were 20 uterine perforations, 12 requiring laparoscopic (seven patients) or laparotomy (five patients) management, three abnormal uterine bleeding, one cervical laceration during dilatation, five fluid overload syndrome and one patient developed intrauterine synechiae (Appendix A).

## 7. Risk Bias Assessment

*Uterine septum versus controls (no septum)—comparative studies:* the quality of the manuscripts was high in one paper [10]; three manuscripts were of medium quality [26,31,33]; and six paper were judged as having low quality [14,27,28,29,30,32]. See Table 4.

*Treated uterine septum versus controls (untreated septum)—comparative studies:* the quality of manuscript was high in one study [19]; seven manuscripts were of medium quality [29,36,37,38,39]; and one paper were judged as having low quality [40]. See Table 5.

*Before and after septum removal—non comparative studies:* none of the papers were judged as having high quality. Seven manuscripts had medium quality [12,49,50,52,53,54,58]. The remaining twelve papers were judged as having low quality [41,42,43,44,45,46,47,48,51,55,56,57]. See Table 6.

Table 4, Table 5 and Table 6 quality assessment of studies though methodological index for non-randomized studies (MINORS) [59].

## 8. Discussion

The septate uterus represents a clinical dilemma for the physician who is treating the patient affected with this enigmatic congenital uterine anomaly. Over the last decade, different studies have evaluated the correlation between uterine septum/subseptum (with/without hysteroscopic metroplasty) and reproductive outcomes (PR, LBR, spontaneous abortion, premature delivery). Unfortunately, a comprehensive updated summary of the current available evidence is missing. Two recent publications on this topic questioned the effectiveness of septum surgical correction to improve the patient’s reproductive outcomes [39,40]. However, the above-mentioned studies conflict with two recently published position papers that reasonably rejected their findings, continuing to propose metroplasty as the treatment of choice in patients with a septate/subseptate uterus and a history of infertility or miscarriages [40,66]. Hence, based on current scientific uncertainties, we decided to perform a systematic review and meta-analysis in order to summarize the current scientific evidence regarding the role of uterine septum/subseptum and its surgical treatment on human reproduction and pregnancy outcome.

## 9. Main Findings and Implications

### 9.1. Uterine Septum versus Controls (No Septum)

Regarding the effects of uterine septum on reproductive outcomes (uterine septum versus controls), we found that SA (first and second trimester) and PL were more frequent in patients with uterine septum compared to those without septum; instead, PR and LBR were higher in patients with uterine septum compared to those without septum. However, according to the analysed outcome and our results, some considerations are crucial.

First, based on our analysis, we can affirm that the presence of a uterine septum could have a detrimental effect, especially on LBR, SA. These data were confirmed both on general analysis, subgroup analysis (infertile patients and patients with history of recurrent miscarriage) and sensitivity analysis (excluding low quality papers) with low heterogeneity. This trend was confirmed also in the case of PR and PL in general analysis. At subgroup analysis and sensitivity analysis, these data were not confirmed for PR and PL, even if the low number of included studies and the high heterogeneity precluded generating meaningful conclusions.

To the best of our knowledge, for the first time in a meta-analysis on this topic, detrimental effects on reproductive outcomes were confirmed also in women with uterine sub-septum for SA, LBR and PL. Regarding PR, even if our results were close to significance, the low number of patients with a uterine sub-septum did not allow for the obtaining of meaningful conclusions. Unfortunately, no data could be extracted about the role of the uterine sub-septum length even if we found no statistically significative differences of PR, LBR, spontaneous abortion and preterm delivery comparing patients with uterine septum versus sub-septum.

Our data are in line with two previously published systematic reviews of Venetis et al. and Chan et al. [13,67]. In the former study, the authors, although not distinguishing between partial and complete uterine septum, demonstrated the negative impact of a septate uterus on the ability to maintain pregnancy [13]. In the second, the authors established that the prevalence of uterine anomalies was higher in women with a history of miscarriage compared to the general population [67].

Different to previously published systematic reviews, we also focused our attention on specific subpopulations distinguishing, where possible, the modality of conception (spontaneous versus assisted reproductive technology), the population studied (women with a history of infertility versus recurrent miscarriage), and the trimester of the pregnancy in which the abortion occurred (first versus second trimester abortion).

When we analyzed the SA rate in patient with history of infertility versus recurrent miscarriage, it seems that the detrimental effect of the septum is slightly higher in the population of infertile patients than in those with recurrent abortions (OR 7.56 vs 2.29, respectively). We hypothesize that this finding could be due to the wide range of factors involved in the pathogenesis of recurrent miscarriage, like genetic and chromosomal abnormalities [68], infectious agents [69], hormonal causes, environmental toxicities and oxidative stress [70].

In our analysis we found a clear association between uterine septum and the increased probability of spontaneous abortion both in the first and second trimesters; however, the correlation was higher in the first trimester. This was highlighted in a paper published by Zlopaša et al. [28], who found a percentage of spontaneous abortion of 77.3 % during the first trimester versus 15.9% during the second trimester of pregnancy; in the same way, Saravelos et al. found that the percentage was 72.6 % versus 13.2 % during the first and second trimester of pregnancy, respectively [32].

### 9.2. Treated Uterine Septum versus Untreated Septum

Analyzing the nine papers that compared treated versus untreated uterine septum, we found that hysteroscopic metroplasty could significantly improve LBR on general analysis. This effect was enhanced considering medium/high quality studies. This data was not confirmed in the subgroup analysis (infertile versus recurrent miscarriage). Removal of the septum by hysteroscopy (hysteroscopic metroplasty) would appear to improve the outcome SA, being very close to statistical significance. Indeed, excluding low quality papers, SA was confirmed to be significantly lower in patients who had their uterine septum removed versus unremoved.

In terms of PR and PL, the hysteroscopic correction of septum would seem not to significantly improve the outcomes both in general analysis and in subgroup/sensitivity analysis. These results are in disagreement with previously reported data. However, we ignore if the lack of improvement of PR and preterm delivery rate after surgery could be related to a real inefficiency of the treatment or to the inconsistency of the trials currently available in the literature (high heterogeneity, poor number of high-quality studies). Indeed, Pang et al., one of the few prospective studies, demonstrated that women with a history of recurrent spontaneous abortion who underwent hysteroscopic metroplasty had higher rates of pregnancy and full-term delivery, and lower rates of spontaneous abortion and preterm delivery than women without metroplasty [37]. Conversely, in the recently published paper by Rikken et al. the hysteroscopic metroplasty did not improve reproductive outcomes compared to expectant management in terms of pregnancy rate, pregnancy loss and preterm birth [39]. It is important to highlight that in reference to this paper, a recent letter/position paper that underlined possible biases has been published; the authors strongly criticized the large number of centers from which the data were obtained, the inclusion in the study of patients not infertile at the time of enrollment, the lack of information about other causes of infertility, and the inclusion of multiple operators among other scientific mishaps (probably with heterogeneous techniques of hysteroscopic septum resection).

Recently, the first multicentric randomized trial on this topic was published (TRUST—The Randomised Uterine Septum Trial) [22]. As previously reported, it was methodologically impossible to aggregate this data in our meta-analysis. In this trial, the authors found no significant difference in live birth rate in patients allocated to septum resection versus expectant management (31% versus 15%—relative risk—95% CI 0.47 to 1.65). We certainly recognize the importance of this paper. However, we disagree with the final statement of the manuscript “in light of the lack of any evidence of effectiveness and the potential for harm, we recommend against septum resection as a routine procedure in clinical practice”.

First, we must reassure the patients that office based hysteroscopic septum resection without cervical dilatation could be considered a safe procedure that can be performed without anesthesia or with local anesthesia, with a low rate of only minor complications. [71,72].

Secondly, we must underline the different limitations of this trial. As suggested also by Vercellini et al. the number of patients included and the stated effect size were too low and too high, respectively, to identify smaller but still clinically relevant differences between groups [73]. Another limitation is due to the type of patients included; data from patients with a history of subfertility, pregnancy loss or preterm birth were analyzed together, thus limiting the applicability of the results to a different subset of patients. The variability of septum definition and the non-uniformity of the diagnostic technique applied is another important limiting factor which can lead to significant selection bias. Another problem related to this trial’s methodology is that if the outcomes were estimated 12 months after randomization (septum diagnosis), it is not clear if the follow-up was the same both in the expectant group and treatment group. Finally, the majority of patients showed a partial septum (without specifying septum length), so generalizing these results to the complete septum is in our opinion incorrect. In conclusion, we recognize the merits of this first randomized trial but, unfortunately, in our opinion, the quality of evidence presented is not adequate to change clinical practice based on decades of scientific literature.

In the 2018, the Cochrane Systematic Review Hysteroscopy was published about treating subfertility associated with suspected major uterine cavity abnormalities. Only two randomised studies were included, one concerning the hysteroscopic removal of submucous fibroids, the second about the hysteroscopic removal of endometrial polyps, and no randomised study with regard to the uterine septum. Therefore, we are unable to compare our results. The authors concluded that more research is needed to measure the effectiveness of the hysteroscopic treatment of suspected major uterine cavity abnormalities in women with unexplained subfertility [74].

A meta-analysis on this topic concluded that septum resection was associated with a lower rate of miscarriage compared with untreated women and no significant effect was seen on live birth, clinical pregnancy rate or preterm delivery [75]. Although interesting, this meta-analysis is incomplete considering the poor number of papers included (only seven) and the fact that all patients were evaluated together without distinction between infertile patients and patients with recurrent miscarriage. Moreover, no meta-regression analysis nor subgroup analysis was performed, thus limiting the clinical applicability of the results.

More recently, a systematic review and meta-analysis comparing reproductive outcomes between women undergoing hysteroscopic resection of the uterine septum and those with expectant management supported that hysteroscopic metroplasty was effective in reducing the risk of miscarriage in patients with complete or partial uterine septum. This study was similar with our results, but they only considered treated uterine septum versus untreated septum without evaluating the uterine septum versus controls (no septum) [76].

### 9.3. Before and after Septum Removal

In the last section of our meta-analysis, we included all observational non-comparatives studies that reported reproductive outcomes before and after septum removal in the same subset of the population. This part of the meta-analysis includes most of the papers, this type of trial being easier to perform than comparative observational studies. Interestingly, in this population we found a significant improvement post-resection in both LBR, SA rate and preterm delivery rate. This data was confirmed both in general analysis and also in subgroup analysis (infertile versus recurrent miscarriage) and sensitivity analysis (excluding low quality papers). The most recent revision of a before/after design studies dates from the year 2000 and is in agreement with our data [77]. Nevertheless, we recommend executing caution when interpreting the results from our last section due to the high heterogeneity of included papers, the presence of publication bias and the intrinsic limits related to before/after non-experimental study design.

## 10. Surgical Complications

The total complication rate reported ranged between 1.5% and 1.8%. Of these, the most common complication was uterine perforation (25 events) with or without abdominal surgical correction, followed by eight cases of abnormal bleeding and only six cases of fluid overload syndrome. Concerning uterine perforation, it is interesting to underline that of the 25 cases, 10 cases were reported by one study [12] that used a 26 French operative hysteroscope for metroplasty. This is a fact of considerable importance because the time range of the included studies is large, and so many of them were performed using what now would be considered obsolete medical equipment like the 26 French operative hysteroscope or monopolar energy that could explain the higher complication rate. Currently, recent advances in hysteroscopic technology have created miniaturized safer to use instruments with a diameter of less than 16 French and bipolar energy that are associated with a lower complication rate [71,72].

## 11. Biological Rationale

The definitive explanation for the correlation between the septate uterus and impaired reproductive potential is still unclear, although several biological mechanisms have been hypothesized. Accumulating evidence suggested that one of the main pathophysiological noxa could be represented by a different expression pattern of cytokines and inflammatory mediators in the endometrium surrounding the septum which that may lead to altered endometrial receptivity [20,21]. This theory could explain the results of similar impaired reproductive outcomes in patients with either uterine septum or sub-septum [78], supporting that the implantation in patients with uterine septum presented an alteration in the vascularization pattern of the decidua basalis. Histological analyses of the septum found the presence of more muscular tissue than fibroelastic connective tissue [79], therefore an increased content of muscle tissue could generate an increased and uncoordinated uterine contractility [20]. Furthermore, the endometrium located over the septum presented a reduced number of glandular ostia, irregular non-ciliated cells with rare microvilli, and a decreased ratio of the ciliated to non-ciliated cells [78,80]. This may imply a decrease in the response to preovulatory hormonal changes of the endometrium covering the uterine septum which could play a role in the pathogenesis of primary infertility [81,82]. Recent advances also demonstrated a lower expression profile of HOXA genes in the endometrium of patients with a septate uterus; this may be of paramount importance considering that different studies underlined the important role of HOXA genes for the proper development of the female genital tract, including the endometrium and for endometrial receptivity. [21].

Finally, during the second trimester of pregnancy, the space requirements of the developing fetus increase, so the risk of spontaneous abortion and preterm birth could be associated to the reduction of the intrauterine space in patients with a septate uterus [13].

## 12. Strengths and Limitations

This meta-analysis critically evaluated all of the current available literature about the uterine septum that has been published over the last 20 years, representing the best available evidence on the implications and the management of septate uterus, a topic that is currently highly controversial. Different from previous systematic reviews and meta-analyses [13,67], we focused our attention on the septate/subseptate uterus in order to provide updated answers to clinically relevant questions on the implications of septate uterus and the potential benefit of hysteroscopic treatment of the uterine septa by using a strictly scientific evidence-based approach. For the first time in the literature, we included data from both comparative (septum versus no septum and treated versus untreated septum) and non-comparative studies (before and after septum removal) providing useful information from different patient populations (recurrent abortion and infertile patients). However, results from before-and-after study designs may have some limitations. We cannot exclude that some other influential events, which could affect the outcome, occurred after the intervention during the follow-up; moreover, a change in outcome measure might be explained by a group with a one-time extreme value naturally changing towards a normal value and, finally, the placebo effect cannot be eliminated.

In the sections “Treated uterine septum versus controls (untreated septum)” and “Before and after septum removal” clinical heterogeneity within and between studies represents a limit of this meta-analysis that is worth highlighting. A random-effects model was used for the calculations to adjust for this heterogeneity. The meta-regression analyses supply the information about the potential moderating effect of some characteristics on the observed effect sizes. In most cases, different moderators such as septum classification, mode of conception, types of studies or quality studies did not change the effets size. Through sub-group analysis we assessed for the first time the role of a partial septum. Despite that fact that the evidence was limited by the poor number of studies included, we confirmed worse reproductive outcomes in this subgroup also.

The quality assessment conducted for the purpose of this meta-analysis suggested that most of the studies were of average quality, with only three out of 25 studies being graded as of high quality. Whether the methodological problems of the individual studies have biased the results cannot be assessed, although it might be considered reassuring that, in most cases, the effect size evaluated based on the quality of the studies did not materially alter the conclusions drawn. Finally, the present meta-analysis is also limited by including predominantly observational studies.

## 13. Conclusions and Future Research

The available evidence revealed a correlation between women having a septate uterus and poorer reproductive and obstetrical outcomes (pregnancy rate, live birth rate, miscarriage rate and pre-term delivery rate) compared to women without a septate uterus. Septum resection may be effective in increasing the live birth rate and reducing the risk of spontaneous abortion in women with poor reproductive histories. Non-conclusive evidence can be extrapolated regarding the efficacy of surgery in improving the likelihood of pregnancy in the infertile population. Moreover, the impact of septum resection on the prevention of preterm labor cannot be determined.

Pending further evidence, metroplasty should still be considered as good clinical practice in infertile patients or in patients with a history of repeated spontaneous abortion. [6,83]

In case of a newly diagnosed uterine septum in patients seeking pregnancy without a prior history of reproductive failure, any advantages from septum removal cannot be established due to a lack of clinical scientific data. Further research through well designed comparative studies and randomized trials are needed to draw conclusions about the best management of patients diagnosed with a septate uterus.

## Figures and Tables

**Figure 1 jcm-11-03290-f001:**
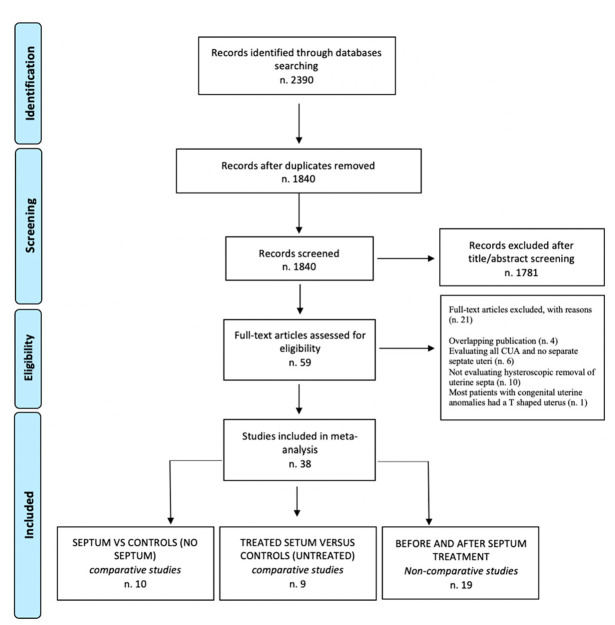
Flowchart summarizing literature identification and selection [65].

**Figure 2 jcm-11-03290-f002:**
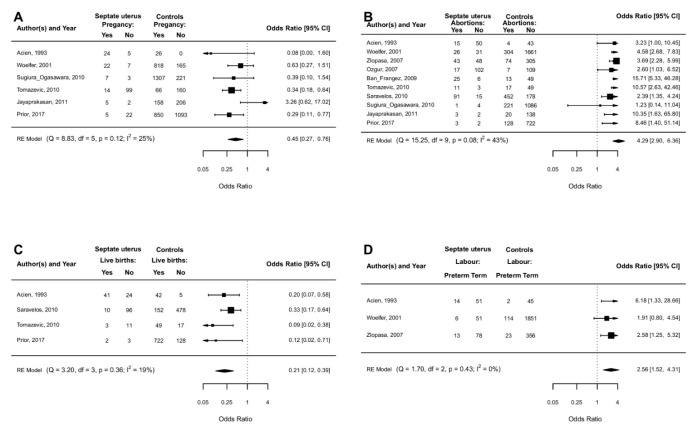
Septate uterus vs. controls (no septate uterus): clinical pregnancy rate (**A**), spontaneous abortions in I-II trimesters (**B**), live birth rate (**C**), and preterm labour (**D**).

**Figure 3 jcm-11-03290-f003:**
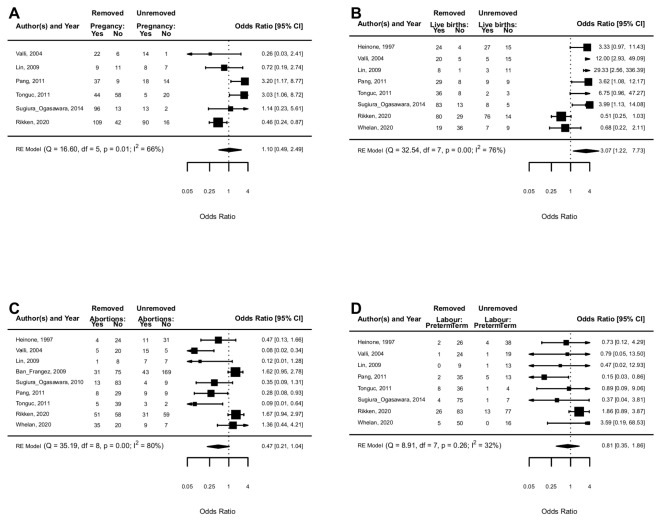
Treated uterine septum vs untreated uterine septum: clinical pregnancy rate (**A**), live birth rate (**B**), spontaneous abortions in I-II trimesters (**C**), and preterm labour (**D**).

**Figure 4 jcm-11-03290-f004:**
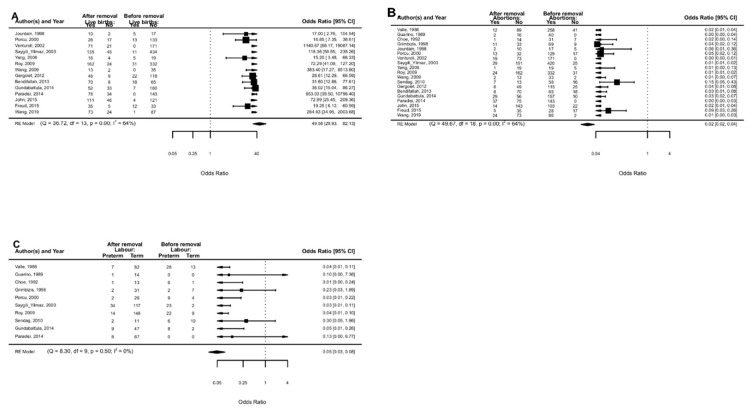
Before vs. after septum removal: live birth rate (**A**), spontaneous abortions in I-II trimesters (**B**), and preterm labour (**C**).

**Table 1 jcm-11-03290-t001:** General features of studies included in the meta-analysis: Uterine septum versus controls (no septum).

Study	Design	Period	Population	Exclusion Criteria	Cases	Controls	Sample Size; Case/Control	Outcomes	Follow-Up	Mode of Diagnosis	Method of Classification
Acien et al., 1993, [14]	Retrospective	1980–1991	Infertile: if attempts to achieve pregnancy remained unsuccessful for >two years. Recurrent miscarriage Natural conception	Patients with: Rokitansky syndrome, hypoplastic, arcuate, bicornuate, didelphis, unicornuate uterus.	Septate uterus Subseptate uterus	No malformations.	31/28	Pregnancy rate Live birth rate Spontaneous abortion Preterm labour	24 months	Clinical examination, ultrasound hysterosalpingographypyelography.	Jarcho (1946) [24], Buttram and Gibbons (1979) [25] and the AFS (1988) criteria [5]
Woelfer et al., 2001, [26]	Prospective	Aug 1997 to Sep 2000	Infertile Recurrent miscarriage Natural conception	Women ongoing pregnancy, history of infertility or recurrent miscarriage, presence of uterine fibroids that distorted the uterine cavity.	Subseptate uterus	No malformations.	29/983	Pregnancy rate Spontaneous abortion Preterm labour	-	3D TV-US	AFS (1988) criteria [5]
Ozgur et al., 2007, [27]	Retrospective	Jan 2002 to June 2004	Infertile: Male factor Tubal factor Other female factor Unexplained Multifactorial ART conception: IVF	Not reported.	Septate uterus	No malformations.	119/116	Spontaneous abortion	-	2D TV-US HSC	N/A
Zlopaša et al., 2007, [28]	Retrospective	1997–2000	Recurrent miscarriage: experienced at least two early spontaneous abortions Natural conception	Twin gestations, submucosal myomas, fetal chromosomopathy, IVF; arcuate bicornuate, didelphys, unicornuate uterus.	Septate uterus Subseptate uterus	No malformations.	31/182	Spontaneous abortion Preterm labour	-	Surgery hysterosalpingograph sonohysterography laparoscopy HSC	AFS (1988) criteria [5]
Ban-Frangez et al., 2009, [29]	Retrospective	1993–2004.	Infertile: Tubal factor Endometriosis Endocrinologic Male factor Idiopathic ART conception: IVF or ICSI	Extrauterine pregnancies, multiple pregnancies and cases with an empty gestational sac.	Septate uterus	No malformations.	106/212	Spontaneous abortion	-	2D TV-US HSC	AFS (1988) criteria [5]
Sugiura-Ogasawara et al., 2010, [30]	Retrospective	1986–2007.	Recurrent miscarriage: two or more (two–12) consecutive miscarriages whose subsequent pregnancies Natural conception	Patients with chromosome abnormalities, unicornuate, bicornuate, didelphis uterus.	Septate uterus	No malformations.	10/1528	Pregnancy rate Spontaneous abortion	-	Laparoscopy Laparotomy MRI	AFS (1988) criteria [5]
Tomazevic et al., 2010, [31]	Retrospective	1993–2005	Infertile: Tubal factor Endometriosis Endocrinologic Male factor Idiopathic ART conception: IVF or ICSI	Not reported.	Septate uterus	No malformations.	289 embryo transfers, 1654 embryo transfers as controls.	Pregnancy rate Live birth rate Spontaneous abortion	-	2D TV-US HSC	AFS (1988) criteria [5]
Saravelos et al., 2010, [32]	Retrospective	N/A	Recurrent miscarriage: was defined as three or more consecutive pregnancy losses prior to 24 weeks of gestation Natural conception	Pregnancies in which patients received medical treatment or surgery.	Septate uterus	No malformations.	29/107	Live birth rate Spontaneous abortion	-	2D TV-US Hysterosalpingography	AFS (1988) criteria [5]
Jayaprakasan et al., 2011, [33]	Prospective	2005–2009	Infertile: Tubal factor Endometriosis Endocrinologic Male factor Idiopathic ART conception: IVF or ICSI	Patients found to have one or more uterine fibroids or polyps distorting the endometrial cavity.	Septate uterus	No malformations.	7/364	Pregnancy rate Spontaneous abortion	-	3D TV-US	AFS (1988) criteria [5]
Prior et al., 2017, [10]	Prospective	May 2009 to November 2015	Infertile: Tubal factor Endometriosis Endocrinologic Male factor Idiopathic ART conception: IVF or ICSI	Patients with a bicornuate uterus Not possible to make a definitive diagnosis for limited views, distortion of endometrial cavity by fibroids, and intrauterine contraceptive device in situ.	Septate uterus Subseptate uterus	No malformations.	23/1943	Pregnancy rate Live birth rate Spontaneous abortion	Present but not specified	3D TV-US	AFS (1988) criteria [5]

HSC: Hysteroscopy. TV-US: transvaginal-ultrasound. MRI: Magnetic Resonance Imaging.

**Table 2 jcm-11-03290-t002:** General features of studies included in the meta-analysis: 2. Treated uterine septum versus controls (untreated septum).

Study	Design	Period	Population	Exclusion Criteria	Sample Size; Case/Control	Outcomes	Cases	Controls	Follow-Up	Type of Hysetroscopy	Procedure/Metroplasty	Mode of Diagnosis	Method of Classification
Heinonen et al., 1997, [34]	Retrospective	1962–1995	Women who experienced recurrent abortions and/or infertility. Natural conception Recurrent miscarriage: one or more miscarriages	N/A	19/19	Pregnancy rate Spontaneous abortion Preterm labour Live birth rate	Women with uterine septum and subseptum who underwent hysteroscopic metroplasty. Septate uterus Subseptate uterus	Septate and subseptate with no treatment matched by age, gravidity and type of uterine anomaly.	Mean nine years (nine months—25 years)	26-gauge resectoscope (Karl Storz).	Jones procedure and Tompkins procedure respectively in six and 14 cases.	Not reported.	Buttram and Gibbons (1979) [25] criteria
Valli et al., 2004, [35]	Prospective	1990–2001.	Women who experienced recurrent abortions. Recurrent miscarriage: at least two miscarriages. Natural conception	Women with bicornuate uterus.	28/15	Pregnancy rate Spontaneous abortion Preterm labour Live birth rate	Women with uterine septum who underwent hysteroscopic metroplasty. Septate uterus	Untreated women.	36 months	Hamou hysteroscope (Karl Storz).	Resectoscope loop. Septum incision was performed until the underlying myometrial tissue.	HSC	AFS (1988) criteria [5]
Ban-Frangez et al., 2009, [29]	Retrospective	1993–2004.	Women with uterine septum that underwent IVF or ICSI and had a singleton pregnancy (fetal heart activity by ultrasound demonstration) Infertile: Tubal factor Endometriosis Endocrinologic Male factor Idiopathic ART conception: IVF or ICSI	Extrauterine pregnancies, multiple pregnancies and cases with an empty gestational sac.	106/212	Spontaneous abortion Preterm labour	Women who had conceived after IVF or ICSI. Septate uterus	Women without uterine anomalies with a singleton pregnancy after IVF or ICSI	-	N/A	N/A	2D TV-US HSC	AFS (1988) criteria [5]
Lin et al., 2009, [36]	Retrospective	1998–2007.	Complete septum (from fundus to vagina). Infertile: defined as unsuccessfully trying to conceive for at least 1 full year Recurrent miscarriage: at least two miscarriages. Natural conception	Other abnormalities than utero- cervico-vaginal septum.	21/15	Pregnancy rate Spontaneous abortion Preterm labour Live birth rate	Vaginal septum removal plus hysteroscopic resection of the uterus septum. Septate uterus	Women that remained untreated.	Mean 18 months (six months—nine years)	N/A	N/A	3D TV-US hysterosalpingography	N/A
Pang et al., 2011, [37]	Prospective	January 2006 to March 2011.	Women with subseptate uterus and who have experienced recurrent spontaneous abortions (two first- trimester abortions). Recurrent miscarriage: at least two first-trimester Natural conception	Only one spontaneous abortion.	46/32	Spontaneous abortion Preterm labour Live birth rate	Women with uterine subseptum and who have experienced recurrent spontaneous abortion. Subseptate uterus	Women with subseptate uterus and who have experienced recurrent spontaneous abortion not treated with hysteroscopic metroplasty.	15 months	N/A	N/A	3D TV-US	AFS (1988) criteria [5]
Tonguc et al., 2011, [19]	Retrospective	January 2006 to January 2009.	Patients with a uterine septum and otherwise unexplained infertility. Infertile Natural conception	Patients who had a history of tuberculosis or endometriosis, endocrinologic problem, abdominal surgery, husband with mild or severe oligospermia at the spermiogram.	102/25	Pregnancy rate Spontaneous abortion Preterm labour Live birth rate	Women with uterine septum who underwent hysteroscopic metroplasty. Septate uterus	Patients who rejected the surgery.	14 months	26-Fr rigid hysteroscope (Karl Storz).	Incision of the septa at the lower margin and continued upward with a horizontal section from one tubal ostium to the other.	N/A	AFS (1988) criteria [5]
Sugiura-Ogasawara et al., 2014, [38]	Prospective	January 2003 to June 2009.	Women with a history of two or more consecutive miscarriages or one stillbirth and septate uterus. Recurrent miscarriage: at least two consecutive miscarriages Natural conception	Patients with a bicornuate uterus.	109/15	Pregnancy rate Spontaneous abortion Preterm labour Live birth rate	Women with uterine septum who underwent hysteroscopic metroplasty. Septate uterus	Patients who rejected the surgery.	Present but Not specified	N/A	Transcervical resection (TCR) or a Jones modified metroplasty.	Hysterosalpingography and/or 2D transvaginal ultrasound were used as the initial screening. Laparoscopy/ laparotomy and/or MRI	AFS (1988) criteria [5]
Rikken et al., 2020, [39]	Retrospective	January 2000 to August 2018.	Women with a history of subfertility, pregnancy loss or preterm birth. Infertile: defined as the inability to conceive for a minimal period of one year of trying to conceive Recurrent miscarriage: two or more, not necessarily consecutive Natural conception	Women that do not have a wish to conceive at time of diagnosis.	151/106	Pregnancy rate Spontaneous abortion Preterm labour Live birth rate	Women with uterine septum who underwent hysteroscopic metroplasty. Septate uterus Subseptate uterus	Women who had expectant management.	Mean 46 months	73 procedures performed with a Versa point device. 32 with scissors. 12 with electro-surgery. 34 unknown.	Intrauterine septum was completely removed.	Hysterosalpingography, 3D TV-US MRI saline or gel infusion sonohysterography or hysteroscopy combined with laparoscopy.	AFS (1988)[5] and ASMR (2016) [6] criteria
Whelan et al., 2020, [40]	Prospective	July 2004 to May 2012	Women with recurrent early pregnancy loss, uterine septum diagnosed and one subsequent pregnancy. Recurrent miscarriage: two or more documented pregnancy losses before 10 weeks of gestation	N/A	21/11	Spontaneous abortion Preterm labour Live birth rate	Women with recurrent early pregnancy loss, uterine septum diagnosed and one subsequent pregnancy who underwent histeroscopic metroplasty. Septate uterus	Women with recurrent early pregnancy loss, uterine septum diagnosed and one subsequent pregnancy untreated.	N/A	N/A	N/A	HSC 3D TV-US	ASMR (2016) criteria [6]

HSC: Hysteroscopy; TV-US: transvaginal-ultrasound; MRI: magnetic resonance imaging.

**Table 3 jcm-11-03290-t003:** General features of studies included in the meta-analysis: Before and after septum removal.

Study	Design	Period	Population	Exclusion Criteria	Simple Size	Case	Outcomes	Follow-Up	Type of Hysteroscopic	Procedure/Metroplasty	Mode of Diagnosis	Method of Classification
Valle et al., 1986, [41]	Retrospective	N/A	Infertile Recurrent miscarriage Natural conception	N/A	124	Women with uterine septum who underwent hysteroscopic metroplasty.	Spontaneous abortion Preterm labour	N/A	N/A	N/A	N/A	N/A
Guarino et al., 1989, [42]	Retrospective	N/A	Infertile Recurrent miscarriage Natural conception	N/A	19	Women with uterine septum who underwent hysteroscopic metroplasty.	Spontaneous abortion Preterm labour	six months	N/A	N/A	N/A	N/A
Choe et al., 1992, [43]	Retrospective	August 1986 to April 1990.	Infertile Recurrent miscarriage Natural conception	N/A	14	Women with uterine septum who underwent hysteroscopic metroplasty.	Spontaneous abortion Preterm labour	N/A	Nd-YAG laser with an 8-mm operating hysteroscope.	An incision was made from one cornua down the septum, then across the septum. Next, beginning at the opposite cornua, the same procedure was done until each tubal ostium could be seen in the panoramic hysteroscopic view.	Laparoscopy HSC	N/A
Grimbizis et al., 1998, [44]	Retrospective	January 1991 to Dec 1996.	Infertile: Tubal factor Endometriosis Endocrinologic Male factor Idiopathic Recurrent miscarriage: two or more previous miscarriages ART conception: IVF or ICSI Natural conception	N/A	57	Septate uterus Subseptate uterus	Spontaneous abortion Preterm labour	34 months	Rigid hysteroscope mounted with a rotatable 9 mm resectoscope.	Resection from the lower margin of the septum and continued upwards with progressive horizontal incisions in the midline until a normal cavity was obtained.	Laparoscopy HSC	AFS (1988) criteria [5]
Jourdain et al., 1998, [45]	Retrospective	1990–1995.	Infertile Recurrent miscarriage: two or more spontaneous first trimester losses or be infertile. Natural conception	N/A	17	Septate uterus	Live birth rate Spontaneous abortion	41 months	Flexible hysteroscope with Nd-YAG laser.	The septum was divided by the laser after exploration of the cavity.	HSC	N/A
Porcu et al., 2000, [46]	Retrospective	Feb 1988 to December 1996.	Recurrent miscarriage Natural conception	N/A	63	Septate uterus	Live birth rate Spontaneous abortion Preterm labour	48 months	55 procedures performed with a 21-Fr resectoscope. Three procedures with endoscopic scissors. Five procedures with Nd-YAG laser.	Electric section performed in 55 cases (87.3%). Section with a pair of endoscopic scissors in three cases (4.8%). A Nd– Yag laser had been used in five cases (9.5%). The method to divide the septum was chosen on an arbitrary decision and not for a specific reason.	Hysterosalpingography TV-US	N/A
Venturoli et al., 2002, [47]	Retrospective	January 1993 to December 1997.	Infertile: unexplained infertility of at least two years duration Recurrent miscarriage: two or more miscarriages in the last three years Natural conception ART conception	No endocrine or other disorders.	141	Septate uterus	Live birth rate Spontaneous abortion	36 ± 19.5 months	Rigid hysteroscope with a 26 gauge resectoscope (Karl Storz)	After visualization of the tubal ostia, the resection was started from the lower margin of the septum and continued upwards with a progressive horizontal incision in the midline.	Laparoscopy HSC	AFS (1988) criteria [5]
Saygili-Yilmaz et al., 2003, [12]	Retrospective	1990–2000.	Infertile: primary infertility who failed to achieve pregnancy for over two years Recurrent miscarriage: two or more miscarriage Natural conception	N/A	361	Septate uterus Subseptate uterus	Live birth rate Spontaneous abortion Preterm labour	36 months	Rigid hysteroscope with 26 gaude resectoscope and specific loop electrode (Karl Storz).	After visualization both of the tubal ostias, the incision of septa was started from the lower margin and continued upward with horizontal section until a normal cavity was obtained and both tubal ostia could be visualized.	Hysterosalpingography	AFS (1988) criteria [5]
Yang et al., 2006, [48]	Retrospective	N/A	Infertile Recurrent miscarriage Natural conception	N/A	46	Septate uterus Subseptate uterus	Live birth rate Spontaneous abortion	10.2 ± 1.3 months	Rigid hysteroscopy with operating channel into which an optic fiber can be inserted, connected with a Nd-YAG laser.	Incision of the septum between the anterior and posterior uterine walls extending up to the fundus rather into the fundal myometrium.	Laparoscopy HSC	N/A
Wang et al., 2009, [49]	Prospective	Sep 2004 to Oct 2007	Infertile: the inability to conceive after 12 months of unprotected intercourse Recurrent miscarriage: two or more previous spontaneous miscarriages Natural conception	N/A	25	Septate uterus	Live birth rate Spontaneous abortion	17.6 ± 5.4 months	27-Fr hysteroresectoscope (Olympus, Hangzhou City, Japan)	The resection was performed between the anterior and posterior uterine walls, extending up to the fundus rather than into the fundal myometrium from the lower margin of the septum	3D TV-US	AFS (1988) criteria [5]
Roy et al., 2011, [50]	Retrospective	January 2000 to June 2008	Infertile Recurrent miscarriage Natural conception	Presence of endocrine disease, uterine myoma, adnexal disease and abnormal semen parameters in the husband	152	Septate uterus Subseptate uterus	Live birth rate Spontaneous abortion Preterm labour	28 months	A 9-mm working element along with sheath and 4-mm 30° telescope (Karl Storz, Germany) equipped with a hysteroscopic monopolar (Collin’s)	The septum was divided in a cephalad direction until both tubal ostia became clearly available. The resection of septa was stopped at the point when hysteroscope could be moved from one cornual end to another without intervening obstruction, and both tubal ostia could be viewed simultaneously	HSC	AFS (1988) criteria [5]
Sendag et al., 2010, [51]	Retrospective	N/A	Infertile Recurrent miscarriage Natural conception	N/A	30	Septate uterus Subseptate uterus	Spontaneous abortion Preterm labour	N/A	N/A	N/A	N/A	N/A
Gergolet et al., 2012, [52]	Prospective	January 2003 to December 2008.	Recurrent miscarriage: one or more miscarriages Natural conception	Anovulatory cycles, polycystic ovary syndrome and those referred to assisted reproduction treatment.	72	Subseptate uterus	Live birth rate Spontaneous abortion	N/A	8-mm Karl Storzmonopolar operative hysteroscope	N/A	2D/3D TV-US HSC	AFS (1988) criteria [5]
Bendifallah et al., 2013, [53]	Retrospective	January 1999 to December 2009.	Infertile: primary infertility for >3 years as defined by inability to conceive. Recurrent miscarriage: at least two consecutive miscarriages before 16 weeks of gestation. Natural conception ART conception: IVF or ICSI	N/A	128	Septate uterus	Live birth rate Spontaneous abortion	38 months	Operative hysteroscope fitted with a monopolar hook (26F resectoscope, 2.9-mm optical lens; Karl Storz GmbH, Tuttlingen, Germany)	the septum was incised equidistant between the anterior and posterior uterine walls, from the lower margin of the septum and continuing upward with progressive horizontal incisions in the midline to the uterine fundus.	HSC	AFS (1988) criteria [5]
Gundabattula et al., 2014, [54]	Retrospective	2003 to 2010.	Infertile: as the inability to conceive aft er 12 months of contraceptive-free intercourse Recurrent miscarriage: 3 or more pregnancy losses before 24 weeks’ gestation Natural conception	N/A	124	Septate uterus Subseptate uterus	Live birth rate Spontaneous abortion Preterm labour	24 months	26 FR resectoscope with a cutting monopolar electrode or a VersaPoint Hysteroscopy system with a spring-type electrode	The septum was divided in an upward direction until both tubal ostia were visualised in the same plane in a panoramic view of the uterine cavity.	TV-US. Hysterosalpingography	N/A
Paradisi et al., 2014, [55]	Retrospective	January 2001 to June 2007.	Infertile Natural conception	Oligo/anovulation and menstrual irregularities	112	Subseptate uterus	Live birth rate Spontaneous abortion Preterm labour	N/A	Equatorial semicircular loop, cutting 0° with monopolar energy.	After visualization of the tubal ostia the resection was started from the lower margin of the septum and continued until the muscular component until a normal cavity was obtained and the hysteroscope could be moved freely from one tubal ostium to the other without any intervening obstruction.	HSC 3D TV-US	AFS (1988) criteria [5]
John et al. 2015, [56]	Retrospective	2006 to 2014.	Infertile Natural conception ART conception: IVF or ICSI	N/A	286	Septate uterus Subseptate uterus	Live birth rate Spontaneous abortion	N/A	Monopolar cautery loop.	N/A	N/A	N/A
Freud et al., 2015, [57]	Retrospective	2004 to 2011.	Recurrent miscarriage Natural conception ART conception: IVF or ICSI	Multiple pregnancies were excluded from the analysis.	28	Septate uterus	Live birth rate Spontaneous abortion	N/A	Bipolar versascope system (Gynecare, Johnson and Johnson, Somerville, NJ, USA)	Septum was divided until complete visualization of both tubal ostia at the same plane was achieved.	HSC 3D TV-US	N/A
Wang et al., 2019, [58]	Retrospective	Jul 2006 to January 2017.	Infertile: defined as the inability to conceive for a minimal period of 1 year of trying to conceive Recurrent miscarriage Natural conception	Pelvic lesions, such as endometriosisoligo or anovulation and menstrual irregularities; partners with abnormal semen analysis.	121	Septate uterus Subseptate uterus	Live birth rate Spontaneous abortion	12	7 mm rigid hysteroscope (Karl Storz).	Electrosurgical incision in the uterine septum was made equidistantly between the anterior and posterior uterine walls and went up high into the uterine fundus until the presence of a normal-shaped cavity was obtained.	Laparoscopy HSC	AFS (1988) [5] and ASMR (2016) [6] criteria

HSC: Hysteroscopy; TV-US: transvaginal-ultrasound.

**Table 4 jcm-11-03290-t004:** Uterine septum versus controls (no septum)—comparative studies.

Study	A Clearly Stated Aim	Inclusion of Consecutive Patients	Prospective Collection of Data	Endpoints Appropriate to the Aim of the Study	Unbiased Assessment of the Study Endpoint	Follow-Up Period Appropriate to the Aim of the Study	Loss to Follow Up less than 5%	Prospective Calculation of the Study Size	An Adequate Control Group	Contemporary Groups	Baseline Equivalence of Groups	Adequate Statistical Analyses	Final Score
Acien et al., 1993. [14]	**	**		**		**	*		*	**	*	*	14
Woelfer et al., 2001. [26]	**	**	**	**	*			*	**	**	**	**	18
Ozgur et al., 2007. [27]	**	**		**					**	**	**	**	14
Zlopaša et al., 2007. [28]	**	*	*	**					*	**	**	**	13
Ban-Frangez et al., 2009. [29]	**	**		**					**	**	**	**	14
Sugiura-Ogasawara et al., [30]	**	**		**		*	*		**	**	**	**	16
Tomazevic et al., 2010. [31]	**	**	*	**					**	**	**	**	15
Saravelos et al., 2010. [32]	**	**	**	**	*				**	**	**	**	17
Jayaprakasan et al., 2011. [33]	**	**	**	**	*	*		*	**	**	**	**	19
Prior et al., 2017. [10]	**	**	**	**	*	*	**	**	**	**	**	**	22

Each item is scored as “missing” not reported, “*” reported but inadequate; “**” reported and adequate.

**Table 5 jcm-11-03290-t005:** Treated uterine septum versus controls (untreated septum)—comparative studies.

Study and Year	A clearly stated Aim	Inclusion of Consecutive Patients	Prospective Collection of Data	Endpoints Appropriate to the Aim of the Study	Unbiased Assessment of the Study Endpoint	Follow-Up Period Appropriate to the Aim of the Study	Loss to Follow Up less than 5%	Prospective Calculation of the Study Size	An Adequate Control Group	Contemporary Groups	Baseline Equivalence of Groups	Adequate Statistical Analyses	Final Score
Heinonen et al., 1997 [34]	**	*	**	**	*	**		**	*	**	*	**	18
Valli et al., 2004 [35]	**	**		**	*	**	**	*	**	**	**	**	20
Ban-Frangez et al., 2009 [29]	**	**	**	**	*			**	**	**	**	**	19
Lin et al., 2009 [36]	**	*	**	*	*	**	**	**	*	**	**	**	20
Pang et al., 2011 [37]	**	**	**	**	*	**		**	**	**	*	**	20
Tonguc et al., 2011 [19]	**	**	**	**	**	**		*	**	**	**	**	21
Sugiura-Ogasawara et al., 2014 [38]	**	**	**	**	**	*		*	*	**		**	17
Rikken et al., 2020 [39]	**	*	**	**	*	**	*	*	**	**		**	18
Whelan et al., 2020 [40]	**	*	*	**	*	*			*	**	*	**	13

Each item is scored as “missing” not reported, “*” reported but inadequate; “**” reported and adequate.

**Table 6 jcm-11-03290-t006:** Before and after septum removal—non comparative studies.

Study and Year	A Clearly Stated Aim	Inclusion of Consecutive Patients	Prospective Collection of Data	Endpoints Appropriate to the Aim of the Study	Unbiased Assessment of the Study Endpoint	Follow-Up Period Appropriate to the Aim of the Study	Loss to Follow Up less than 5%	Prospective Calculation of the Study Size	Final Score
Valle et al., 1986 [41]	**	*		*					4
Guarino et al., 1989 [42]	**	*		**					5
Choe et al., 1992 [43]	**			*		*	*		5
Grimbizis et al., 1998 [44]	**	*		**		**	*		8
Jourdain et al., 1998 [45]	**		*	*		**	*		7
Porcu et al., 2000 [46]	**			**		**	*		7
Venturoli et al., 2002 [47]	**	*		**		*	**		8
Saygili-Yilmaz et al., 2003 [12]	**	**		**		**	*		9
Yang et al., 2006 [48]	**	*		**		*	*		7
Wang et al., 2009 [49]	**	**		**		**	*		9
Roy et al., 2011 [50]	**	**		**		**	**		10
Sendag et al., 2010 [51]	**			**					4
Gergolet et al., 2012 [52]	**	**	**	**	*			*	10
Bendifallah et al., 2013 [53]	**	**		**		**	*		9
Gundabattula et al., 2014 [54]	**	**		**		*	**		9
Paradisi et al., 2014 [55]	**	**	*	**		*			8
John et al., 2015 [56]	**	**		*					5
Freud et al., 2015 [57]	**	**		**					6
Wang et al., 2019 [58]	**	**	**	*		**	*		10

Each item is scored as “missing” not reported, “*” reported but inadequate; “**” reported and adequate.

## Data Availability

Upon request.

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
