# Peer review of "Uterine Septum with or without Hysteroscopic Metroplasty: Impact on Fertility and Obstetrical Outcomes—A Systematic Review and Meta-Analysis of Observational Research"

_jcm, 2022, doi:10.3390/jcm11123290_

Round 1
Reviewer 1 Report
Comments to the Author
Comment
Many reports have described clinical outcomes after hysteroscopic metroplasty before.In recent years, a paper that septum resection does not improve reproductive outcomes was published in Rikken et al and Krishnan et al. Your manuscript it is very fine to mention the limitation of the paper by Rikken et al and Krishnan et al.
And it is very wonderful that the analysis is divided into infertility and recurrent pregnancy loss in the sub-analysis.
Also, I think it is wonderful to consider risk of bias using MINORS.
However some minor revision is necessary. And I have some comments.
Page 1 of 38
ABSTRACT
Line24
“Preterm labour” is misspelled. The correct spelling is preterm labor.
Page 8 of 38
3.1.2 Sensitivity analysis
In six studies with medium / high quality, CPR was higher after vs. before the removal of the uterine septum….
Is CPR the clinical pregnancy rate?
Is it different from PR?
Page 13 of 38
DISCUSSION
Main Findings and Implications
Uterine septum versus controls (no septum)
Line3
PR and LPR were higher in patients with uterine septum compared to those without septum.
Isn't it LBR instead of LPR?
Page 14 of 38
Line3
In terms of PR and preterm delivery, the hysteroscopic correction of septum…
Do you dare to use preterm delivery instead of preterm labor?
Author Response
ANSWERS TO REVIEWER
Manuscript ID number:
jcm-1731504
Title of paper:
UTERINE SEPTUM WITH OR WITHOUT HYSTEROSCOPIC METROPLASTY: IMPACT ON FERTILITY AND OBSTETRICAL OUTCOMES. A SYSTEMATIC REVIEW AND META-ANALYSIS OF OBSERVATIONAL RESEARCH
Dear Reviewer:
Thank you for giving us the chance to enhance our manuscript “UTERINE SEPTUM WITH OR WITHOUT HYSTEROSCOPIC METROPLASTY: IMPACT ON FERTILITY AND OBSTETRICAL OUTCOMES. A SYSTEMATIC REVIEW AND META-ANALYSIS OF OBSERVATIONAL RESEARCH”
Comment#1
Many reports have described clinical outcomes after hysteroscopic metroplasty before. In recent years, a paper that septum resection does not improve reproductive outcomes was published in Rikken et al and Krishnan et al. Your manuscript it is very fine to mention the limitation of the paper by Rikken et al and Krishnan et al.
And it is very wonderful that the analysis is divided into infertility and recurrent pregnancy loss in the sub-analysis.
Also, I think it is wonderful to consider risk of bias using MINORS.
However, some minor revision is necessary. And I have some comments.
Response: Dear Reviewer, thank you very much for your positive comments.
Comment#2
Page 1 of 38
ABSTRACT
Line24
“Preterm labour” is misspelled. The correct spelling is preterm labor.
Response: Dear reviewer, thank you for your comment. We corrected with “preterm labor”.
Comment#3
Page 8 of 38
3.1.2 Sensitivity analysis
In six studies with medium / high quality, CPR was higher after vs. before the removal of the uterine septum….
Is CPR the clinical pregnancy rate?
Is it different from PR?
Response: Dear reviewer, thank you for your comment. We corrected with “PR”, pregnancy rate
Comment#4
Page 13 of 38
DISCUSSION
Main Findings and Implications
Uterine septum versus controls (no septum)
Line3
PR and LPR were higher in patients with uterine septum compared to those without septum.
Isn't it LBR instead of LPR?
Response: Dear reviewer, thank you for your comment. We corrected in “LBR”
Comment#5
Page 14 of 38
Line3
In terms of PR and preterm delivery, the hysteroscopic correction of septum…
Do you dare to use preterm delivery instead of preterm labor?
Response: Dear reviewer, thank you for your comment. We corrected in “preterm labor”

Reviewer 2 Report
I have read the manuscript and in my opinion it gives very good overview and critical reinterpretation of results.
The introduction well describes the clinical dilemma and aim of the meta-analysis.
The methods are described in detail.
Results are also presented in details including quality assessments of included studies.
Discussion is well written so the reader can find some clinically useful ideas when septum resection is justified and when not.
Overall, in my opinion the manuscript is well written. I've found one typo on page 13, line 3 (LPR instead of LBR).
Author Response
ANSWERS TO REVIEWER
Manuscript ID number:
jcm-1731504
Title of paper:
UTERINE SEPTUM WITH OR WITHOUT HYSTEROSCOPIC METROPLASTY: IMPACT ON FERTILITY AND OBSTETRICAL OUTCOMES. A SYSTEMATIC REVIEW AND META-ANALYSIS OF OBSERVATIONAL RESEARCH
Dear Reviewer:
Thank you for giving us the chance to enhance our manuscript “UTERINE SEPTUM WITH OR WITHOUT HYSTEROSCOPIC METROPLASTY: IMPACT ON FERTILITY AND OBSTETRICAL OUTCOMES. A SYSTEMATIC REVIEW AND META-ANALYSIS OF OBSERVATIONAL RESEARCH”
Comment#1
I have read the manuscript and in my opinion it gives very good overview and critical reinterpretation of results.
The introduction well describes the clinical dilemma and aim of the meta-analysis.
The methods are described in detail.
Results are also presented in details including quality assessments of included studies.
Discussion is well written so the reader can find some clinically useful ideas when septum resection is justified and when not.
Response: Dear Reviewer, thank you for your positive comment for our manuscript.
Comment#2
Overall, in my opinion the manuscript is well written. I've found one typo on page 13, line 3 (LPR instead of LBR).
Response: Dear reviewer, thank you for your comment. We corrected in “LBR”
